# On the distortions in calculated GW parameters during slanted atmospheric soundings

Alejandro de la Torre[1], Peter Alexander[2], Torsten Schmidt[3], Pablo Llamedo[1], Rodrigo Hierro[1]

*Correspondence to*: Alejandro de la Torre (adelatorre@austral.edu.ar)

[1] Facultad de Ingeniería, Universidad Austral and CONICET, Mariano Acosta 1611, Pilar, Provincia de Buenos Aires B1629ODT, Argentina

[2] IFIBA, CONICET, Ciudad Universitaria, C1428EGA, Buenos Aires, Argentina

[3] GFZ, GFZ German Research Centre for Geosciences, Section 1.1: GPS/Galileo
Earth Observation, Telegrafenberg A17, D-14473 Potsdam, Germany

**Abstract.** The significant distortions introduced in the measured atmospheric gravity wavelengths by soundings other than in vertical and horizontal directions, are discussed as a function of elevation angle of the sounding path and the gravity waves aspect ratio. Under- or overestimation of real vertical wavelengths during the measurement process depends basically on the value of these two parameters. The consequences of these distortions on the calculation of the energy and vertical flux of horizontal momentum are analyzed and discussed in the context of two experimental limb satellite setups: GPS-LEO radio occultations and TIMED/SABER measurements. Possible discrepancies previously found between the momentum flux calculated from satellite temperature profiles, on site and from model simulations, may, to a certain degree, be attributed to these distortions. A recalculation of previous momentum flux climatologies based on these considerations seems to be a difficult goal.

## 1. Introduction

In the last few years, we have observed the ongoing development of several techniques to sound the lower, middle and upper atmosphere (e.g., Wu and Waters, 1996; Tsuda et al., 2000; Preusse et al., 2002; S.P. Alexander, et al., 2011; Hertzog et al., 2012; John and Kumar, 2013; Lieberman et al., 2013; Oliver et al., 2013; M.J. Alexander, 2015; de Wit et al., 2017). The advantages and disadvantages of each choice are clearly distinguishable among the available rocket-, balloon-, satellite- and satellite-borne instruments, as well as radar and lidar ground-based devices. Regarding the retrieval of information on atmospheric dynamics from satellite measurements, we know that both satellite limb and nadir observing techniques are needed to resolve different parts of the gravity wave (GW) spectrum (Wu et al., 2006) and that a better understanding of

GW complexities requires joint analyses of these data and high-resolution model simulations. The global observation of the atmosphere and the ionosphere using limb or nadir sounding paths, makes it possible to obtain vertical profiles of refractivity, density, temperature ($T$), pressure, water vapor content and electron density, which is a remarkable achievement obtained with the available experimental resources.

One of the main objectives pursued by current observations is the permanent improvement required in the understanding of GW sources of generation (such as flow over topography, convection, and jet imbalance), as well as their propagation, breaking and dissipation around and above the tropopause, forcing atmosphere circulation. This knowledge, is needed in the sub-grid parameterizations in global models for climate and weather forecasting applications, in order to simulate the influence of orographic and non-orographic GWs and produce realistic winds and temperatures (e.g., Fritts and Alexander, 2003; McLandress and Scinocca, 2005; Kawatani et al., 2009; M.J. Alexander et al., 2010; Shutts and Vosper, 2011; Geller et al, 2013). In these parameterizations, some parameters describe the global distributions of GW vertical flux of horizontal momentum ($MF$), as well as their wavelengths and frequencies. Until recently, the necessary parameters could not be determined through global observations because the waves are small in scale and intermittent in occurrence. The parameterizations compute a momentum forcing term by making assumptions about the unresolved wave properties that have not been properly constrained by observations. The assumptions are formulated as a set of tuning parameters that are used to adjust the circulation and temperature structure in the upper troposphere and middle atmosphere (M.J. Alexander et al., 2010).

Among recently developed sounding devices, Global Positioning System (GPS) Radio Occultation (RO) is a well-established technique for obtaining global GW activity information. RO uses GPS signals received by Low Earth-Orbiting (LEO) satellites for atmospheric limb sounding. $T$ profiles are derived with high vertical resolution and provide a global coverage under any weather conditions, offering the possibility to carry out the global monitoring of the vertical $T$ structure and atmospheric wave parameters. Several authors have contributed to global analyses of horizontal and vertical GW wavelengths, specific potential energy and $MF$ distribution (Tsuda et al., 2000; de la Torre et al., 2006; Wang and Alexander, 2010; Faber et al., 2013, Schmidt et al. 2016; M.J. Alexander et al. 2015). In particular, P. Alexander et al (2008) (A08) stated that it is not possible to fully resolve GW from RO measurements because there are different kinds of distortions. In each occultation, the outcome depends on wave characteristics (essentially wavelengths and amplitude), the line of sight (LOS) and the line of tangent points (LTP), both with respect to the phase fronts to be detected. Ideal conditions for accurate wave amplitude extraction in occultation retrievals are given by quasi horizontal wave phase surfaces or when the LOS and LTP are respectively nearly contained and out of those planes. Short horizontal scale waves are weakened or even filtered out with high probability. Another result from A08 is that the detected vertical wavelengths will always differ from the original ones, but only the presence of inertio-GWs, which have nearly horizontal constant phase surfaces, will ensure small discrepancies. They concluded that extreme caution is needed when addressing the issues of amplitude, wavelength and phase of gravity waves in occultation data. Some years before A08, de la Torre and P. Alexander (1995) (TA95) already observed and established analytically the discrepancies to be expected between measured and real

horizontal and vertical wavelengths during balloon soundings, taking into account the motion of the gondola with respect to the constant GW phase surfaces. This analysis was performed both from the intrinsic and the ground frame of reference.

In Sect. 2, we analyze in general the distortion to be expected in the detection of real vertical and horizontal wavelengths from almost instantaneous soundings that are different from vertical and horizontal, specifically for satellite measurements. In Sect. 3, the consequences of this distortion in the calculation of GW energy and *MF* are discussed. In Sect. 4, the situation for two satellite setups is considered in some detail. In Sect. 5, some conclusions are outlined for future applications and a possible careful reconsideration of some results and conclusions obtained in previous climatologies is suggested.

**2. GW wavelengths distortion**

From TA95 and A08, it is clear that when an on site or remote sensing instrument sounds the atmosphere along a given direction, which is different from the vertical or the horizontal plane, the measured vertical and horizontal wavelengths are expected to considerably differ from "real" (or "actual") values. In the Appendix from TA95, 1) a stationary GW observed from 2) a ground-fixed frame of reference (Figure A1 and Eqs. A1-A5) was specifically considered. Now, it may be accepted that both these conditions are emulated by GPS-LEO RO (e.g., Kursinski et al., 1997), as well as by TIMED/SABER (Atmosphere using Broadband Emission Radiometry/Thermosphere-Ionosphere-Mesosphere-Energetics and Dynamics) (Russell et al., 1999) measurements (see below Sect. 4). In relation to the first condition, we may assume that satellite-based soundings yield *T* profiles almost instantaneously. Following this reasoning, the vertical "real" and "apparent" (or measured) wavelengths ($\lambda_z$ and $\lambda_z^{ap}$ respectively) are related by the following expression (TA95, Eq. A3-A5):

$$\lambda_Z^{ap} = \frac{\lambda_Z}{abs(1+\cot(\alpha)\cot(\psi))} \tag{1}$$

where $\alpha$ is the elevation angle defined by a straight sounding path direction and the horizontal plane. In turn, cot ($\psi$) is the ratio between the horizontal wavenumber vector ($k_H$) projected on the vertical $\alpha$ - plane, and the vertical wavenumber $k_Z$ (Figure 1). The ratio ($k_H / k_Z$) is also known as the GW aspect ratio. Fig. 1, with two arbitrary successive GW phase surfaces $\phi_1$ and $\phi_2$ cutting $\alpha$-the plane defined, shows a clear difference between real and apparent vertical (and horizontal) wavelengths. This distortion, frequently present in for example radiosoundings or satellite-based GW studies, is in general non negligible and affects the calculation of all magnitudes requiring previous identification of wave parameters.

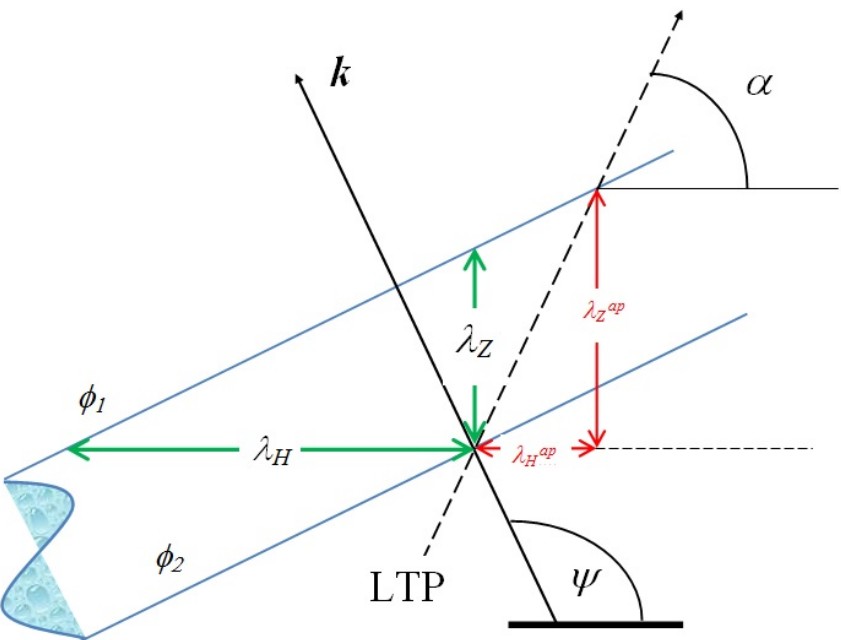

Figure 1: Vertical $\alpha$-plane defined by the elevation angle between the sounding path direction and its projection on the horizontal plane. Real and apparent vertical and horizontal wavelengths are indicated. $\phi_1$ and
$\phi_2$ represent two arbitrary successive constant phase surfaces belonging to a monochromatic GW (see text). The expected distortion from real to apparent (or measured) wavelengths, is seen.

Here we recall that $\cot(\alpha)$ is equal to the ratio between $\lambda_H^{ap}$ and $\lambda_Z^{ap}$; this result will be used below. A similar relation to Eq. (1) may be derived between horizontal real and apparent wavelengths, from Eq. A3 to A6 in TA95. The resulting relation is (not shown in TA95):

$$\lambda_H^{ap} = \frac{\lambda_H}{abs(1+\tan(\alpha)\tan(\psi))} \qquad (2)$$

We should mention that $\lambda_H$ is real but may not be the true horizontal wavelength, as information must be sampled along two different horizontal directions, at least to be able to calculate it (e.g. Faber et al 2013, Schmidt et al 2016). We will now focus on the consequences derived from the expected distortion in $k_Z$ or in $\lambda_Z$. As is known, in global atmospheric models the subgrid parameterization of GW energy and $MF$ is based on a successful identification of GW parameters, after proper processing of $T$ profiles. The effects of GW on the large-scale circulation have been treated via parametrizations in both climate and weather forecasting applications. In these parametrizations, key parameters describe a global distribution of $MF$, GW wavelengths and frequencies (e.g. Alexander et al., 2010).

Eq. (1) provides the magnitude of the expected departure in $\lambda_Z^{ap}$ from $\lambda_Z$, for each monochromatic GW component, within a given wave ensemble at any atmospheric region. In order to better understand this distortion, we will consider this equation as parametric in $\alpha$ or $\psi$. We recall that, as stated above, both independent parameters are simple trigonometric functions of the apparent and real (horizontal/vertical) wavenumber components ratio, respectively. The angle $\alpha$ only depends on the sounding path direction during

the observation process through progressive atmospheric layers, and $\psi$, on the GW direction of propagation, $\mathbf{k}$ $/ k$. Here, $\mathbf{k}$ and $k$ are the wavenumber vector and its absolute value, respectively. We note here that Eq. (1) is symmetric with respect to $\alpha$ and $\psi$, which are in turn, totally unrelated. For example, in the case of GPS-LEO RO measurements (to be considered below in Sect. 4), $\alpha$ represents the angle defined by the Line of Tangent Points (LTP) and the horizontal plane. In Fig. 1, an arbitrary segment of LTP is roughly represented by a

straight line. In this figure we observe, for example, that a vertical sounding of the atmosphere in the nadir direction (i.e., lidar measurements or balloon measurements under zero wind conditions) will produce no distortion at all in $k_Z$ or in $\lambda_Z$. The same can be said for horizontal soundings producing no distortions in $k_H$ or in $\lambda_H$ belonging to the $\alpha$-plane.

In Fig. 2, we define the distortion as the ratio:

$$D = \frac{\lambda_Z^{ap}}{\lambda_Z} \qquad\qquad (3)$$

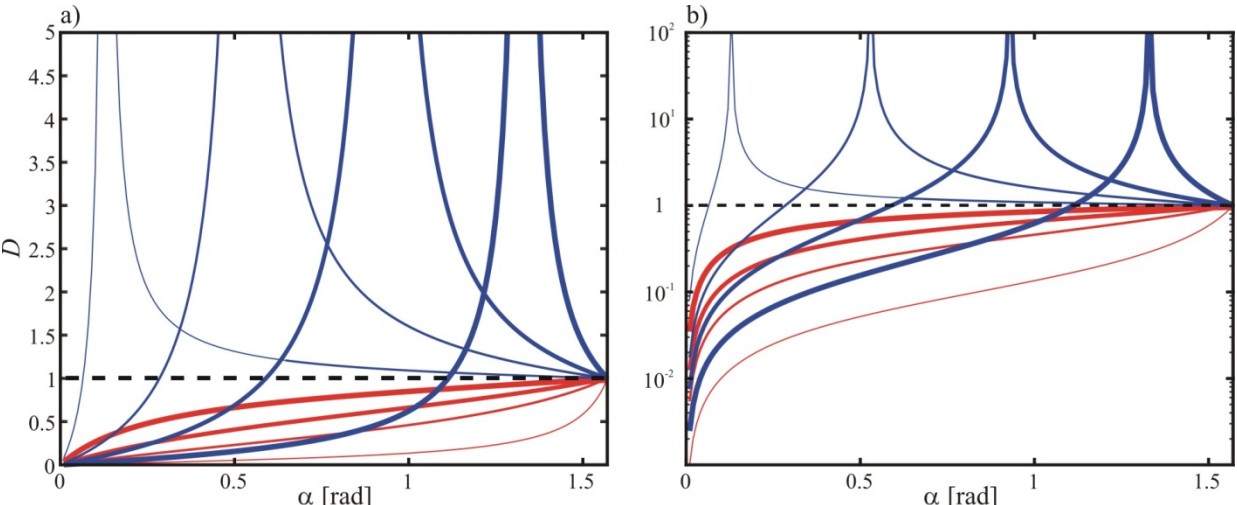

Figure 2: (a) Examples of the distortion in $D = \lambda_Z^{ap} / \lambda_Z$, as a function of $\alpha$, leaving a parametric dependence with $\psi$ (see text). Arbitrary constant and progressive $\psi$ values within the interval $[0, \pi]$ rad are shown.

Underestimation of $\lambda_Z$ occurs when $(D < 1)$ $\psi = 0.1, 0.5, 0.9, 1.3$ rad (curves with increasing thickness, from

thin red to thickest red, respectively). Overestimation of $\lambda_Z$ occurs when $D > 1$ and $\psi = 1.7, 2.1, 2.5$ and $2.9$ rad (thin blue to thickest blue curves, respectively). Note that the upper blue branches for $\psi > \pi/2$ diverge at

singular values. This is better appreciated in Fig. 2b. The horizontal dashed line corresponds to the "non distortion" case. Considerable general departures from this non distortion limit ($D = 1$) are seen. (b) The same content as in (a), here in linear-log axes.


Following Eq. (1), $D$ may be equivalently represented as a function of $\alpha$ leaving $\psi$ as a parameter, or vice versa, making use of the symmetric dependence on both of them. We first describe this function in terms of $\alpha$ in Fig. 2a and 2b. For illustration, we show the variation of $D$ for increasing selected values of $\psi$ between 0 and $\pi$ rad. Note that the underestimation of $\lambda_Z$ occurs when ($D < 1$) $\psi = 0.1$, 0.5, 0.9, 1.3 rad and the

overestimation of $\lambda_Z$ occurs when ($D > 1$) $\psi = 1.7$, 2.1, 2.5 and 2.9 rad. For each $\psi$ value, a singular $\alpha$ value associated to two upper diverging branches is seen. This is better appreciated in Fig. 2b. The horizontal dashed line corresponds to the "non distortion" $D = 1$ case. Considerable general departures from this non-distortion limit are seen. Note that the functional behavior of $D$ is non-symmetric for $\psi$ greater than and less than $\pi/2$ rad. Also, notice that all possible sounding and wave orientations are covered by defining one of the

angles between 0 and $\pi/2$ rad and the other one between 0 and $\pi$ rad.

From the above arguments, we can conclude that for a given GW ensemble, a net significant distortion of the measured spectra should be expected. This net distortion will become more or less significant, depending on i) the composition of the ensemble and ii) the specific measuring device. In the next section we will illustrate this argument for the case of satellite-borne measurements. A 3D plot presents better the functional

dependence of $D$ with $\psi$ and $\alpha$ already shown in Fig. 2a-b, now separately for under- and overestimations of $\lambda_Z$, below and above the plane $D = 1$ (Fig. 3a and 3b respectively).

a)                                                    b)

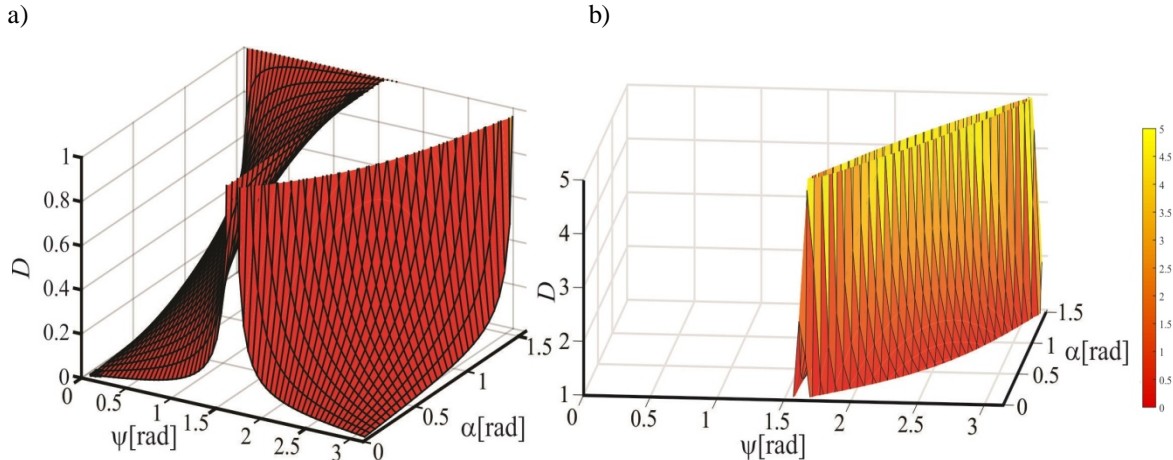

Figure 3: (a) (left) 3D perspective of the distortion $D$ already shown in Figure 2, as a function of $\psi$ and $\alpha$ (see text), for underestimations of $\lambda_Z$. (b) (right) The same as in (a), for overestimations of $\lambda_Z$. Black lines in this

figure are only intended to make easier a visual appreciation of the curved nature of the surfaces. The color bar illustrates $D$ values in both plots for intervals [0,1] and [1,5] respectively.

The 3D plot shows the complete variability of $D$ for $a$ between 0 and $\pi/2$ and $\psi$ between 0 and $\pi$. For any fixed $\psi$ value, starting at $\alpha = 0$, each $D$ curve increases from zero, crosses the $D = 1$ boundary diverging at a given $\alpha$ value, located (after/before) $\pi/2$ depending on $\psi$ is (less/greater) than $\pi/2$ and decreases again to zero, as $\alpha$ approaches the $\pi$ limit. Due to the symmetric dependence of $D$ with both parameters, to avoid a possible confusion and redundancy, in Fig. 2 it seems enough to show the $D$ variability for $\alpha$ between 0 and
$\pi/2$.

### 3. GW energy, spectra and momentum flux

The computation of the specific potential energy per unit mass, $Ep$, for a GW ensemble, is given by:

$$Ep = \frac{1}{2}\left(\frac{g}{N}\right)^2 \overline{\left(\frac{\hat{T}}{T_0}\right)^2} = \frac{1}{2}\left(\frac{g}{N}\right)^2 \frac{1}{z_2-z_1}\int_{Z_1}^{Z_2}\left(\frac{\hat{T}}{T_0}\right)^2 dz \qquad (4)$$

Where $z_1$ and $z_2$ are the minimum and maximum altitudes for integration, $g$ is the acceleration due to gravity, $N$ is the buoyancy frequency, $\hat{T}$ and $T_0$ are the perturbation amplitude and background temperature, respectively, and the overbar indicates a space averaging process. This average must be performed, for the GW ensemble considered, over at least one wavelength corresponding to the GW mode with the largest amplitude in any direction (i.e., horizontal, slanted or, as usually, vertical). Consistently, different choices of
this direction involving the same ensemble should ideally yield identical results. Alternatively, the average may be also performed over a time interval at a fixed point, considering a general non-stationary ensemble of GW. In this case, the net contribution of stationary waves would be obviously underestimated. In addition, we recall that the computation of instantaneous $Ep$ at fixed points is sometimes reported without the corresponding averaging process, but we consider that this procedure lacks a clear physical sense.

In Eq. (4) we must previously remove noise and long scale structures in the $T$ profiles. The remaining GWs should include amplitudes expected to significantly contribute to $Ep$. The vertical interval for integration is usually about 10 km. But, depending on $\alpha$, $\psi_i$ and the azimuth of each one of the dominant modes in the GW ensemble, some waves may not be contained at least for one complete cycle within the integration interval. Then, the integral in Eq. (4) may not include at least one full wavelength from all these dominant modes. As a
result, the individual contribution of each mode to the net $Ep$ will be under- or overestimated to a significant extent.

To extend these considerations to a quite realistic scenario, let us consider a particular modelled distribution of GW vertical wavelengths, selected among the numerous theories developed and based on diverse

experimental setups, after the seminal paper by Dewan and Good (1986) (e.g., Smith et al., 1987; Hines, 1991; Fritts and Alexander, 2003; Yiğit et al., 2017, and references therein). It has been observed and broadly assumed, that part of a GW spectrum (the larger vertical wavenumbers) is saturated beyond a given characteristic $k_Z^C$ value that decreases with increasing altitude. Smaller wavenumbers than $k_Z^c$ are not expected to be saturated and their amplitudes increase with increasing altitude. One example of the spectral models proposed to describe the energy density, $E$, assumes its separability in the product of three functions $A$, $B$ and $C$, depending respectively on the vertical wave number, the intrinsic frequency, $\omega$, and the azimuthal direction of propagation, $\Phi$ (Fritts and VanZandt, 1993):

$$Ep(k_Z, \omega, \Phi) = A(k_Z)B(\omega)C(\Phi) = A_0 \frac{1}{\frac{k_Z^c}{k_Z} + \left(\frac{k_Z}{k_Z^C}\right)^3} B(\omega)C(\Phi) \tag{5}$$

In the above form, $A(k_Z)$ takes into account the requirement of a positive slope (to get a finite vertical energy flux) at small wavenumbers and the proposed $k_Z^{-3}$ dependence at large wavenumber values. This "universal model" has been the subject of several objections and variations in the last three decades (see e.g. Fritts and Alexander, 2003). Note that, a given $Ep$ distribution like Eq. (5) is obtained based on an experimental setup (for example, the parameters may be derived after an analysis of COSMIC GPS RO $T$ data). Consistently, $k_Z$ as well as $k_Z^C$ should then be considered apparent values, estimated after a spectral analysis (e.g., Tsuda et al., 2011). For vertical (i.e. lidar) soundings, apparent and real parameters are indistinguishable. Following this argument, as

$$k_Z^{ap}{}_{(1,2)} = k_{Z\,(1,2)}[abs(1 + \cot(\alpha)\cot(\psi))]^{-1} \tag{6}$$

we consider Eq. (5) with $k_Z^{ap}$ instead of $k_Z$ and $k_Z^{C,ap}$ instead of $k_Z^C$ to quantitatively illustrate the distortion in $Ep$ and (below) in $MF$, derived from the misinterpretation between real and apparent parameters. In doing so, the GW energy contained in a given vertical wavenumber interval $\Delta k_z^{ap}$ is:

$$Ep_{\Delta k_Z}{}^{ap} = A_0 B(\omega)C(\Phi) \int_{k_{Z1}{}^{ap}}^{k_{Z2}{}^{ap}} \frac{1}{\frac{k_Z^{c,ap}}{k_Z^{ap}} + \left(\frac{k_Z^{ap}}{k_Z^{c,ap}}\right)^3} dk_Z^{ap} =$$

$$A_0 B(\omega)C(\Phi) \left[ \frac{\tan^{-1}\left(\frac{k_{Z2}{}^{ap2}}{k_Z^{c,ap2}}\right)}{2k_Z^{c,ap-1}} - \frac{\tan^{-1}\left(\frac{k_{Z1}{}^{ap2}}{k_Z^{c,ap2}}\right)}{2k_Z^{c,ap-1}} \right] \tag{7}$$

Let us assume that from a given slanted sounding, after extracting the GW perturbations with a wavelet or bandpass filtering analysis, a clearly dominant quasi monochromatic wave packet, encompassed by two apparent wavenumber bounds, $k_{Z1}{}^{ap}$ and $k_{Z2}{}^{ap}$, is identified. We may calculate the wave energy associated to this wave packet, directly from Eq. (7). The relative error in $Ep$ may be estimated after replacing apparent by real wavenumbers in (7). To simplify the argument, we assume in Eq. (6) that $k_{Z1}{}^{C,ap}$ and $k_{Z2}{}^{C,ap}$ are close

enough to assume a parametric dependence with constant $\alpha$ and $\psi$ values. The relative error in *Ep* takes the form:

$$\frac{\Delta Ep}{Ep} = \frac{\left| \left[ \frac{\tan^{-1}\left(\frac{k_{Z2}^{ap2}}{k_Z^{c,ap2}}\right)}{2k_Z^{c,ap-1}} - \frac{\tan^{-1}\left(\frac{k_{Z1}^{ap2}}{k_Z^{c,ap2}}\right)}{2k_Z^{c,ap-1}} \right] - \left[ \frac{\tan^{-1}\left(\frac{k_{Z2}^{2}}{k_Z^{c2}}\right)}{2k_Z^{c-1}} - \frac{\tan^{-1}\left(\frac{k_{Z1}^{2}}{k_Z^{c2}}\right)}{2k_Z^{c-1}} \right] \right|}{\left[ \frac{\tan^{-1}\left(\frac{k_{Z2}^{2}}{k_Z^{c2}}\right)}{2k_Z^{c-1}} - \frac{\tan^{-1}\left(\frac{k_{Z1}^{2}}{k_Z^{c2}}\right)}{2k_Z^{c-1}} \right]} =$$

$$[abs(1 + \cot(\alpha)\cot(\psi))]^{-1} \tag{8}$$

It is to say, under the above assumptions, the relative error in *Ep* does not depend on vertical wavenumbers or parameters other than simply $\alpha$ and $\psi$.

The *MF* for internal GWs may be calculated under certain hypotheses based on the existence of a dominant mode characterized by $\lambda_Z$ and $\lambda_H$ within a given intrinsic frequency range, applying the following equation (for its detailed derivation and discussion refer to Appendix A of Ern et al. (2004)):

$$MF = \frac{\rho}{2} \frac{\lambda_Z}{\lambda_H} \left(\frac{g}{N}\right)^2 \overline{\left(\frac{\hat{T}}{T_0}\right)^2} = \rho \frac{\lambda_Z}{\lambda_H} Ep \tag{9}$$

where $\rho$ is the background density. Note that in this derivation, the dominant mode with $\lambda_Z$ and $\lambda_H$ dominates within the narrow wavenumber interval mentioned above in the discussion of the spectral distribution of *Ep*. A first order estimation of the *MF* relative error may be derived, by propagating up to first order the relative errors in *Ep* and $(\lambda_Z/\lambda_H)$. The relative error in *MF* will simply result in the sum of those relative errors:

$$\frac{\Delta(MF)}{MF} = \left| \frac{\Delta\left(\frac{\lambda_Z}{\lambda_H}\right)}{\frac{\lambda_Z}{\lambda_H}} \right| + \left| \frac{\Delta E_p}{E_p} \right| = \left| \frac{\left(\frac{\lambda_Z}{\lambda_H}\right)^{ap} - \left(\frac{\lambda_Z}{\lambda_H}\right)}{\left(\frac{\lambda_Z}{\lambda_H}\right)} \right| + \left| \frac{\Delta E_p}{E_p} \right|$$

$$= \left| \frac{\tan(\alpha) - \cot(\psi)}{\cot(\psi)} \right| + [abs(1 + \cot(\alpha)\cot(\psi))]^{-1} \tag{10}$$

remembering that $\lambda_Z^{ap}/\lambda_H^{ap} = \tan(\alpha)$ and $\lambda_Z/\lambda_H = \cot(\psi)$. Note that, under the above assumptions, the *MF* relative error does not depend on the wavenumber bounds nor on the wavenumber width of the GW packet considered. Note that an erroneous replacement in Eq. (9) of apparent instead of real wavelengths, would absurdly lead to the conclusion that the *MF* would depend on the geometry of the sounding path. To provide a measure of the distortion in *MF* from data retrieved during a specific slanted case study, let us consider a GPS RO slanted sounding close to Andes mountains analyzed in detail by Hierro et al. (2017) (in what follows,

H17). In that case study, from a collocation database between RO and cloud data and from Weather Research and Forecasting (WRF) mesoscale model simulations, real and apparent vertical wavelengths during COSMIC RO soundings were identified. From the model, coherent bi-dimensional GW structures with constant phase surfaces oriented from SW to NE were noted. From the orographic quasi monochromatic structures detected below the cloud tops, an average $\lambda_Z \approx 22.5$ km and $\lambda_H = 20$ km were estimated, yielding the ratio $\lambda_Z/\lambda_H = 1.12$ with a wave propagation angle $\psi = \tan^{-1}(\lambda_H/\lambda_Z) \approx 0.73$ rad. In this case study, the LOS stands at each TP almost aligned to the GW phase surfaces observed, it is to say, at 190° from north direction (dotted lines in Fig. 7 from H17). This particular geometry between LOS and constant phase surfaces should allow to observe vertical oscillations in the RO profile corresponding to short $\lambda_H$ structures, as described in A08. We recall that in Sec. 2 we mentioned that $\alpha$ may be calculated from a rectilinear approximation of the LTP and cot $(\alpha)$ is also equal to the ratio between $\lambda_H^{ap}$ and $\lambda_Z^{ap}$ in the region and altitude interval considered in H17. From the average inclination of LTP, cot $(\alpha) = \lambda_H^{ap} / \lambda_Z^{ap} \approx 0.68$ rad, which considerably differs from the ratio between the corresponding real wavelengths, $\lambda_H /\lambda_Z = 0.89$. From Eq. (9) the proportionality of $MF$ to the real wavelengths ratio indicates that when this ratio is erroneously replaced by the apparent wavelengths ratio, a significant error is in the general case, introduced.

As stated above, the estimation of the $MF$ relative error for this particular Andes case study results:

$$\frac{\Delta(MF)}{MF} = \left|\frac{\tan(\alpha) - \cot(\psi)}{\cot(\psi)}\right| + [abs(1 + \cot(\alpha)\cot(\psi))]^{-1} = 0.31 + 0.57 = 0.88. \qquad (11)$$

The error result should be observed as indicative, as the uncertainty affecting the determination of the parameters $\alpha$ and $\psi$ should be remembered.

Now we may wonder about the logically expected following point: would the distortion previously described and clearly affecting a single case study, be able to affect the results and conclusions from any specific existing GW global or local climatology? At first glance, given the slanted nature of soundings upon which a given climatology is obtained and the anisotropic nature of the dependence on $\alpha$ and $\psi$, we have no reason to assume that the distortion expected on each sounding should be averaged out in the climatology, notwithstanding the available density of soundings. To try to answer this question, the option to accurately calculate one by one the distortions introduced respectively in each sounding is clearly not possible, due to the unknown $\psi$ parameter. Nevertheless, in an effort to address this point, we resort to one idealized modelled distributions of GW available in the literature (Alexander and Vincent, 2000). This is a linear model describing one-dimensional GW propagation through a vertically varying background atmosphere. It was used to clarify the relationship between GW properties at stratospheric heights and the GW sources at the troposphere. The authors aimed to test whether all of the observational results retrieved from radiosonde profiles could be synthesized into a consistent physical model of a spectrum of vertically propagating GW. In doing so, modelled energy densities and $MF$ were computed before they were compared with the radiosonde

results. The model uses the general dispersion relation for the intrinsic and ground-based frequency, $\hat{\omega}$ and $\omega$ respectively, including a background zonal wind $u$ and Coriolis acceleration $f$, derived i.e. in Gill (1982):

$$\hat{\omega}^2 = (\omega - k_H u)^2 = \frac{N^2 k_H{}^2 + f^2(k_z{}^2 + \mu^2)}{k_H{}^2 + k_z{}^2 + \mu^2} \tag{12}$$

where $N$ is the buoyancy frequency, $\mu = (2H)^{-1}$ and $H$ is the density scale height. The GW source is specified as a distribution of $MF$ versus horizontal phase speed, $c = \omega/k_H$, for fixed $k_H$ values. In this model, the intrinsic frequency and vertical wavenumber vary with $u$ and stability, while $k_H$ remains constant. The

changes in $\hat{\omega}$ with $u(z)$ refer to Doppler shifting and the changes in $k_z$ with $u(z)$ are referred to as refraction (see Alexander and Vincent (2000) for details). From the different GW sources proposed by these authors as spectra of $MF$ versus phase speed located at fixed tropospheric heights, we illustratively consider a source function that is perfectly antisymmetric and isotropic:

$$B_0(c) = B_m\left(\frac{c - u_0}{c_w}\right) exp\left(1 - \left|\frac{c - u_0}{c_w}\right|\right) \tag{13}$$

Where $B_m$ represents a spectral amplitude and $c_w$ a source spectrum width. Note that in the high-middle frequency approximation and neglecting $\mu$, we may write the argument in (13) as:

$$\frac{c - u_0}{c_w} = \frac{\hat{\omega}_i}{k_H c_w} = \left[\frac{N^2 k_H{}^2}{k_H{}^2 + k_z{}^2}\right]^{0.5} \frac{1}{k_H c_w} = \frac{N\,|cos(\psi)|}{k_H c_w} \tag{14}$$

We now analyze the explicit inclusion of the previous distortion $D$ parameter in the scope of this model. As stated, we assume only GW within the high or middle intrinsic frequency regime, neglecting $f$ and $\mu$. The fitting of $MF$ from modelled results ($MF^{mod}$) to measured radiosonde data ($MF^{mea}$) at a fixed location and for

constant $k_H$, involves a comparison between $MF$ profiles which are, in essence, functions of real and apparent data, respectively. Then it looks reasonable to fit of modelled to measured data after applying the corresponding transform to the modelled source spectrum. In doing so, we replace $cos\,\psi$ in Eq. (14) following Eq. (6):

$$D = \frac{k_z}{k_z^{ap}} = abs(1 + cot(\alpha)\cot(\psi)) =$$

$$\begin{cases} 1 + cot(\alpha)\cot(\psi)\,, & if\ 1 + cot(\alpha)\cot(\psi) > 0 \\ -1 - cot(\alpha)\cot(\psi)\,, & if\ 1 + cot(\alpha)\cot(\psi) < 0 \end{cases} \tag{15}$$

In the first case,

$$\psi = cot^{-1}\frac{D - 1}{cot\,\alpha} \tag{16}$$

$$\cos\psi = \cos\cot^{-1}\left(\frac{D-1}{\cot\alpha}\right) = \frac{1}{\sqrt{1 + \left(\frac{\cot\alpha}{D-1}\right)^2}} \qquad (17)$$

after applying a trigonometric identity and for over- or under estimation of $k_Z$, it is to say, when D is different from one. Eq. (13) as a function of $D$, for constant $B_m$, $N$, $k_H$, $c_w$ and viewing path $\alpha$ is (i.e., $\alpha$ is expectedly constant during any radiosounding with uniform and constant background wind):

$$B_0(D) = B_m\left(\frac{N}{k_H c_w}\frac{1}{\sqrt{1 + \left(\frac{\cot\alpha}{D-1}\right)^2}}\right)exp\left(1 - \frac{N}{k_H c_w}\frac{1}{\sqrt{1 + \left(\frac{\cot\alpha}{D-1}\right)^2}}\right) \qquad (18)$$

Finally, under the second case of Eq. (15), $D$-1 is to be replaced by $-D$-1. Following this reasoning, we may expect that this or any other source function, expressed from the onset in terms of measured data that undergo distortions due to the slanted nature of the soundings, will provide, for some optimum value of $D \neq 1$ the best fit to a given experimental $MF^{mea}$ profile. This may provide for example, a quantitative estimation of the distortion to be expected in a climatology at a fixed geographic point. To resume the idea, what really matters

in any quantitative estimation of the distortion introduced by the slanting nature of atmospheric soundings (radiosoundings, radio occultation profiles, etc) is to consistently compare real(apparent) modelled data with real(apparent) measured data.

## 4. Distortion of vertical wavelengths for specific setups

To illustrate the considerations from Sect. 2 and 3, let us consider the $T$ retrievals obtained from 1) RO events detected from different LEO-GPS satellites and from 2) SABER/TIMED measurements. A GPS-LEO RO occurs whenever a transmitting satellite from the global navigation network at an altitude about 20,000 km rises or sets from the standpoint of a LEO receiving satellite at a height of about 800 km and the signal goes across the atmospheric limb. The doppler frequency alteration produced through refraction of the ray by the

Earth's atmosphere in the trajectory between the transmitter and the receiver is detected and then may be converted into slant profiles of diverse variables in the neutral atmosphere and the ionosphere. GPS-LEO RO observations available since 2001 have been broadly used to study global distributions of GW energy and momentum, mainly in the troposphere and the stratosphere (e.g. de la Torre et al., 2006; Alexander et al.2010; Geller et al., 2013; Schmidt et al., 2016). The RO technique is a global limb sounding technique, sensitive

under all weather conditions to GW with small ratios of vertical to horizontal wavelengths (Wu et al., 2006, P. Alexander et al., 2016). The SABER/TIMED limb measurements provided continuous global $T$ data for the latitude range 50◦N–50◦S from the lower stratosphere to the lower thermosphere and represent an unprecedented opportunity for studying in detail the atmospheric waves, in particular GW, as well as their role in lower and upper atmosphere coupling (e.g., Pancheva et al., 2011). The TIMED satellite provides

observations since January 2002. It measures $CO_2$ infrared limb radiance from approximately 20 to 120 km altitude. Kinetic temperature profiles are retrieved over these heights using local thermodynamic equilibrium (LTE) radiative transfer in the stratosphere and lowest part of the mesosphere (up to 60 km) and a full non-LTE inversion in the mesosphere and lower termosphere (i.e., Mertens et al., 2004; Pancheva and Mukhtarov, 2011).

In Fig. 4a and 4b, LTPs corresponding to both setups are illustratively shown, for the higher tropospheric and lower stratospheric regions bounded by 31◦S-37◦S and 66°W-72°W, close to central southern Andes mountains, during January-February 2009.

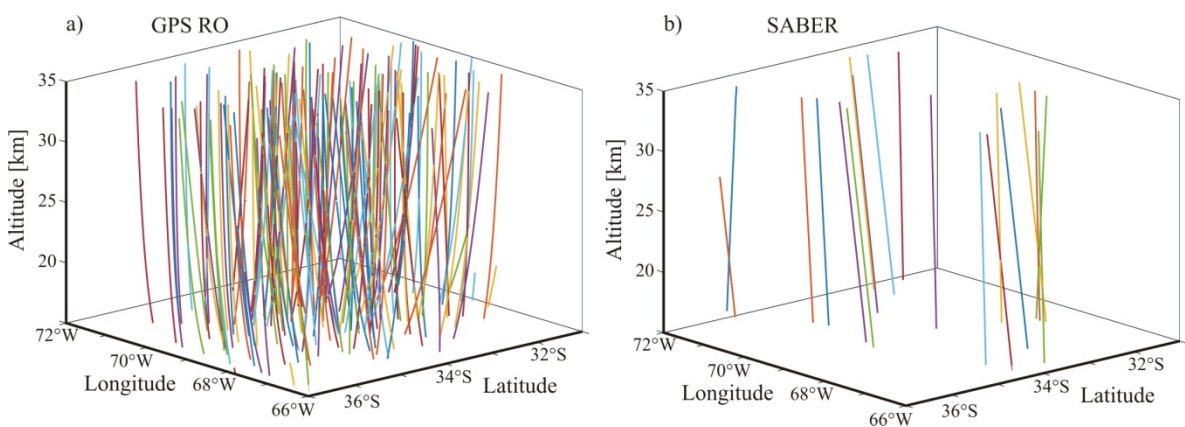


Figure 4: (a) LTPs corresponding to available profiles during the period Jan-Feb 2009, retrieved from tropo-stratospheric GPS-RO events (see text). (b) The same as (a), for SABER measurements. Arbitrary colors were included to make easier the visual inspection.

Keeping in mind that the difference between horizontal and vertical scales in these figures, a typical distribution of the sounding path direction ($\alpha$) among GPS-RO occultation events and among SABER measurements, is observed. The large number of available RO as compared to SABER profiles is evident but no significant variation with latitude was detected. The approximation of the sounding paths by straight segments seems, at least for our purpose here, quite reasonable. Let us now consider the global data retrieved

from both setups during January-February 2009 (RO from LEOs: SAC-C, CHAMP, MetOp-A, and COSMIC), of which Fig. 4 only represents a regional subset. In Fig. 5a and 5b the $\alpha$ distribution is shown. Here a linear interpolation was applied to the weakly variable $\alpha$ angle in each RO event, between the lowest and upper available LTP values. Note the considerably narrower variability $\alpha$-range among SABER profiles. We did not observed remarkable differences in the general latitudinal or geographical distribution. The

possible ranges observed from both experimental setups allow some preliminary consequences to be drawn

regarding the expected wavelength distortions. For example, for the subset in Fig. 4, we know that very close to the Andes mountains region, dominant large-amplitude, stationary and non hydrostatic GWs are usually observed (de la Torre et al., 1996, 2005, 2015).

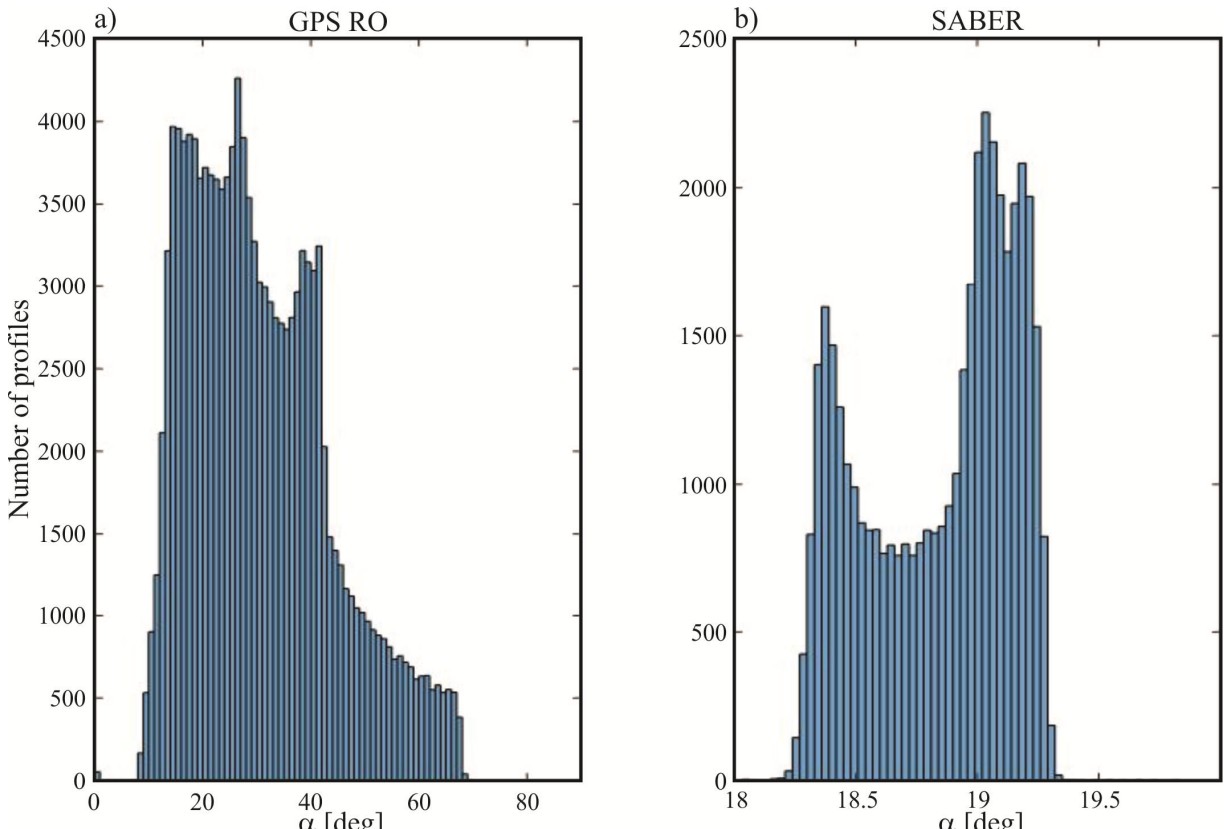


Figure 5: Distribution of available profiles with elevation angle between the sounding path direction and the horizontal plane, globally retrieved, during the period Jan-Feb 2009 from a) GPS-LEO RO and b) TIMED/SABER measurements (see text). The total number of profiles are 127617 and 83712 respectively.


Accordingly, large GW aspect ratios may be expected there (Gill, 1982). On the other hand, at tropical latitudes, where convective GWs dominate the scenario, or even close to polar jet regions where hydrostatic rotating or non rotating GWs are usually found, considerably lower characteristic aspect ratios should be dominant. In Fig. 6, we reproduce the $D$-$\alpha$ curves selected in Fig. 2a and 2b, for successive $\psi$ values ($\Delta\psi$ step = 0.2), now adding in dash-dotted green and yellow squares, the $D$-$\alpha$ ranges affected for both experimental setups. These ranges are, respectively, [0.17-1.22] rad for GPS-RO and [0.32-0.34] rad for SABER. For each

setup, the relevant difference mainly depends on whether $\alpha$ and $\psi$ belong to the same or different $[0, \pi/2]$ and $[\pi/2, \pi]$ intervals.

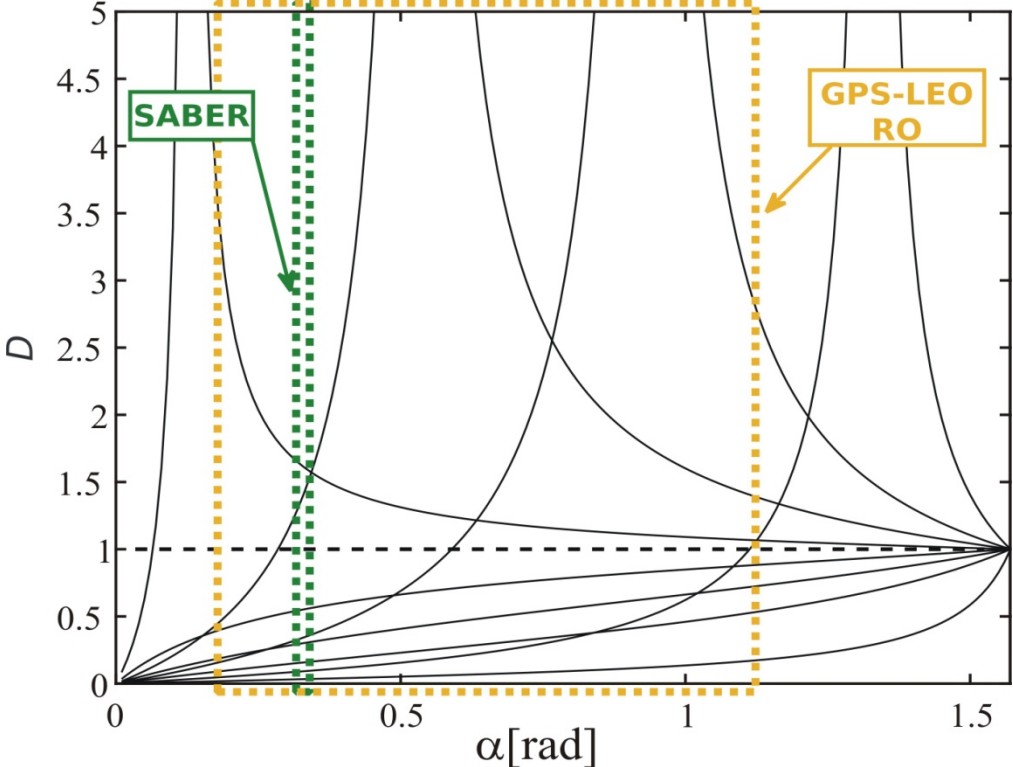

Figure 6: The $\alpha$ ranges corresponding to both experimental setups, are defined within dash-dotted colored boxes. These ranges are [0.17-1.22] rad and [0.32-0.34] rad, respectively in yellow and green, for GPS-RO and SABER, according to Fig. 5. The curves already selected in Fig. 2a and 2b for successive and constant $\psi$ values ($\Delta\psi$ step= 0.4) between 0.1 and 2.9, are included in black for reference purposes.

Here, we may here observe that depending on GW aspect ratio and sounding direction, general under and overestimations of $\lambda_Z$ are both possible throughout both experimental setups. Within a given ensemble, the behavior of $D$ for $\psi$ lower and greater than $\pi/2$ is different. This suggests that different modes in the ensemble may show individual distortions less than or greater than 1. Then, some compensations contributing to $Ep$ and $MF$ are expected from different modes in the ensemble, but the net distortion should still be
considerable. In Fig. 7, the D-$\psi$ constraint imposed to GPS-RO observations, now for constant and progressive $\alpha$ values, is shown. $\Delta\alpha$ steps of 0.02 rad and within the corresponding bounds [0.17-1.22] rad indicated in Fig. 5, are shown. The white, light grey and grey sectors approximately indicate the non-hydrostatic, hydrostatic non-rotating and hydrostatic rotating GW regimes, respectively. We observe general underestimations for $\psi$ less than $\pi/2$ and in the vicinity of $\pi$ rad. Between these sectors, under and
overestimations are possible. To illustrate the consequences on a realistic and simple scenario, let us consider

again the region situated to the east of the central Andes, mentioned in Figures 4a and 4b. Let us suppose that, consistently with observations and numerical simulations (i.e., de la Torre et al., 2012; Jiang et al., 2013; Fritts et al., 2015), constant and stationary GW phase surfaces exhibit a systematic inclination with respect to the ground and a high aspect ratio, following the almost omnipresent forcing by mean westerlies at the

mountain tops. This feature is represented in Fig. 7 by the black arrow.

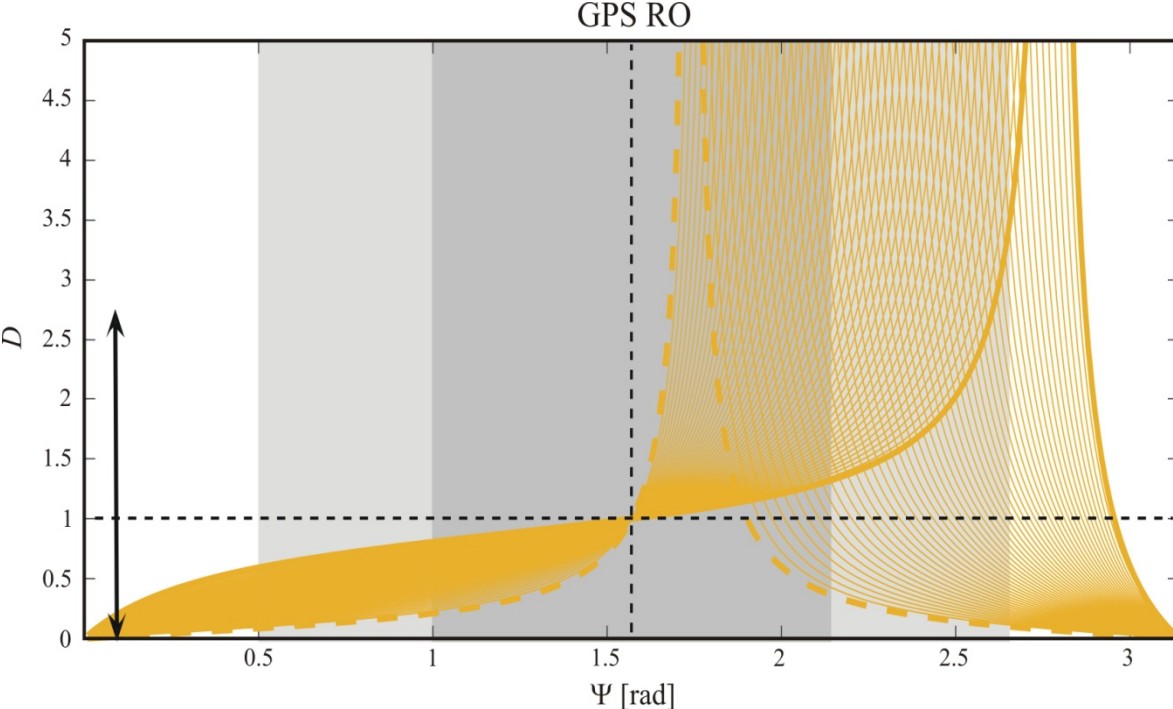

Figure 7: The yellow curves represent progressive and constant selected $\alpha$ values (step $\Delta\alpha = 0.02$ rad). They are selected within the allowed bounds [0.17-1.22] rad in the $D$-$\psi$ GPS RO region, according to Figure 5.

These lower and higher bounds are indicated by thick dotted and full lines, respectively. White, light grey and grey sectors roughly indicate the non-hydrostatic, hydrostatic non-rotating and hydrostatic rotating GW regimes, respectively. Both quadrants are separated by the vertical dashed curve. The black double arrow indicates an hypothetical dominant non-hydrostatic GW that may be observed at different $\alpha$ directions, from different GPS-LEO satellites pairs. The "forbidden GPS-LEO RO sectors" are any sectors excepting those

covered by yellow lines.

This arrow spans over all possible $\alpha$ directions within the bounds imposed by the geometry of every GPS-LEO satellites combination during each occultation event. This assumed scenario would reveal a net underestimation of $\lambda_Z$, regardless of the inclination of LTPs during the sounding of the region and the

considered period. In general the analysis is expected to be more complex, given distinct LTPs contributions

that may under- and overestimate $\lambda_Z$. Finally, Figure 8 indicates the corresponding D-$\psi$ features for SABER measurements, similarly as in Figure 7.

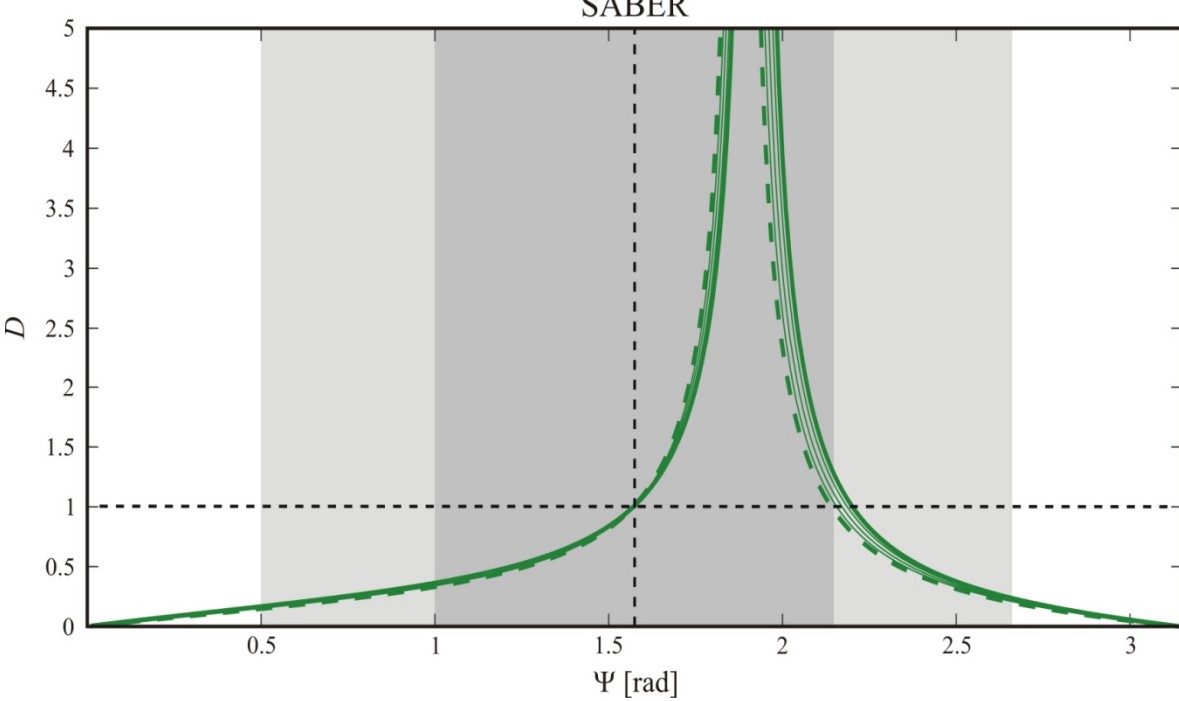

Figure 8: The same as Fig. (7), here for SABER measurements, within the considerably narrower $\alpha$ bounds [0.32-0.34] rad than for the GPS-RO setup, according to Fig. (5). These lower and higher bounds are indicated by thick dotted and full green lines, respectively. The "forbidden SABER sectors" are any sectors excepting those covered by green lines.

Here we observe general underestimations for $\psi$, along the 3 GW regimes, for values less than $\pi/2$ and greater than around 2.3 rad. For intermediate values, only overestimations are expected. Note that for SABER measurements, the forbidden D-$\psi$ region is considerably more extended than for GPS-LEO RO measurements.

## 5. Discussion and conclusions

The expected distortions undergone in the measured vertical wavelengths during any almost instantaneous slanted atmospheric sounding, as may be the case for satellite instruments, is discussed. For the particular case of vertical or horizontal soundings, we know that no distortion is expected in $\lambda_Z$ and $\lambda_{H,}$ respectively. The features observed are described as a function of GW aspect ratio and the inclination of the sounding path.

To gain a better understanding of this distortion, and making use of the symmetric $D$ dependence with $\alpha$ and $\psi$, we consider the expression for $D$ as a parametric equation in both independent variables. To illustrate the constraints imposed to both parameters by applying different instrumental setups and GW scenarios, we show the results conveniently in $D$-$\alpha$ and $D$-$\psi$ plots. Above and below the non-distortion limit ($D = 1$), general under and overestimations occur depending on the relative parametric values. The main difference is produced by two possible situations: $\alpha$ and $\psi$ belonging to the same or different quadrants, taken from [0, $\pi/2$] and [$\pi/2$, $\pi$]. Given a GW ensemble and a number of measurements within arbitrary space bounds and time intervals, distinct wavelength under- and overestimations should be expected.

When $Ep$ is calculated over a GW ensemble in any individual T profile, an integral must be performed over the largest wavelength along any chosen direction. The selection of the upper and lower vertical wavelength bounds, should include those prevailing GW amplitudes expected to mostly contribute to $Ep$. Depending on $\alpha$ and the respective $\psi$ values for each one of the dominant GW modes, some dominant real wavelengths may not be fully contained within the integration interval. The integral in $Ep$ then will not include at least one wavelength of every dominant mode. The $Ep$ calculation could be under- or overestimated up to a significant extent.

We illustrate these arguments in an approximately real scenario considering a modelled distribution of GW. This is based on the usual saturation of large vertical wavenumbers and in the separability of the spectral function in the vertical wave number, the intrinsic frequency and the azimuthal direction of propagation. To calculate the wave energy associated to a given GW packet within an ensemble, we use a simple analytical result derived from the spectral model to get an idea of the distortion expected by wrongly replacing the integration limits by apparent instead of real wavenumber values. This (or any) distortion in $Ep$ will in turn be translated to the $MF$, by applying a previous result obtained by Ern et al. (2004). In addition, through a multiplying factor, the $MF$ would be then illogically dependent on the inclination angle of the sounding path.

The results are considered for two specific experimental setups: GPS-RO and SABER measurements. For our analysis we approximate the sounding paths in both cases by straight segments. The relevance of this assumption was assessed. A clearly larger number of available T profiles is seen from RO events. The $\alpha$ ranges in both techniques allow to define forbidden regions in D-$\alpha$ as well as in D-$\psi$ diagrams, relative to the different GW aspect ratios (the non-hydrostatic, hydrostatic non-rotating and hydrostatic rotating regimes). Within a given GW ensemble, even expecting some compensation when $D$ is less than and greater than 1, the net distortion effect, as well as its contribution to $Ep$ and $MF$, should be considerable. With the exception of GWs with prevailing high aspect ratio, as for example near the Andes mountains where a net underestimation of $\lambda_Z$ should be observed, under- and overestimations are in general expected, from both setups respectively. This occurs for T profiles where $\alpha$ and $\psi$ belong to the same or different quadrants [0, $\pi/2$] and [$\pi/2$, $\pi$]. For SABER measurements, the forbidden D-$\psi$ region is considerably more extended than the one corresponding to the GPS-RO measurements.

In the global study of Geller et al. (2013), which compares models with diverse parameterizations with
satellite and balloon data, the faster fall off with height of the gravity wave *MF* derived from satellite
measurements than in the models considered in that study was the most significant discrepancy between
measured and model fluxes. These authors concluded that the reasons for those differences remain unknown,
although various explanations for the differences were proposed. As we know, from model simulations the
*MF* is not computed from Eq. (8), but from its formal definition, based on the average of the products of the
three perturbed components of the air velocity. Based on the above considerations and regarding the dramatic
distortions on vertical and horizontal wavelengths during slanted soundings, we may infer that if *MF* is
computed from Eq. (8), the wavelengths distortion will unavoidably be translated to the calculation of *MF*.
Obviously, this situation must be considered together with the additional constraints imposed to any satellite-
borne observational window, largely discussed by several authors, including A08. Finally, we must admit that
the global calculation of *MF* from slanted *T* profiles including all necessary corrections, even assuming quasi
monochromatic GW packets, appears to be a very complex task. The distortions described above are only
avoided in the calculation of *MF* if the atmosphere is sounded in the vertical or horizontal directions, as
provided (but only locally) by lidar/radar and balloon setups, respectively. Up to now, from the satellite data
at disposal, an attempt to quantitatively illustrate the implications and possible misrepresentation (or
distortion) of our general understanding of GW parameters values from slanted soundings, as their global
distribution and variability, seems unrealistic. After some research to improve this scenario, we are now
working on previous GW parameters solution schemes which were modified for the use of close sounding
groups of RO profiles. The method is currently being applied to calculate GW propagation direction, net MF
and real vertical and horizontal wavelength for some case studies. The unavoidable constraint imposed to
extend preliminay results to a future GW climatological useful description is strictly conditioned by the still
largely insufficient density of satellite-based soundings.

**Acknowledgements**

The study has been supported by the CONICET under grants CONICET PIP 11220120100034 and ANPCYT
PICT 2013-1097 and by the German Federal Ministry of Education and Research (BMBF) under grant
01DN14001.

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
