# Peer review of "On the distortions in calculated GW parameters during slanted atmospheric soundings"

_Atmospheric Measurement Techniques, 2017_

## Referee Comment (RC1) · P. Pisoft (Referee) · 19 Sep 2017

**General comments**

This is a novel and interesting study of an important aspect of GW parameters distortion due to slanted soundings. The distortion is theoretically derived and specific application is illustrated on GPS-RO and TIMED-SABER observations. Also, this is most probably the first paper that analyzes in details the distortion of GW parameters in relation to the wave front orientation and considering utilized instrument viewing-angle. The manuscript is generally well-written and should be published in AMT.

[Figure]

The paper brings a highly valuable remainder for the scientific community as it describes significant distortions introduced in the measured atmospheric gravity wavelengths by atmospheric soundings that are realized in other than in the vertical or horizontal directions. Resulting under or overestimation of real GW vertical and horizontal wavelengths is projected into calculated GW parameters or related spectral analysis. That may have significant consequences for comparison of GW parameters from different sources and general understanding of the parameters characteristics.

**Specific comments**

As for the specific comments, I have little to say as my previous comments were taken into account after the first review – the questions regarding saturated spectra and pseudo-momentum flux were answered (see below Reactions from the quick review), part regarding quantification of the distortion using a specific data was incorporated into the paper. I have only a suggestion for the discussion part connected to the quantification of the GW parameters distortion. Authors have already provided example of the quantification but it would be useful to elaborate this more also in the discussion to illustrate the implications and possible misrepresentation and distortion of our general understanding of GW parameters values, their (global) distribution, variability etc.

**Technical corrections**

p5/l123 - wavenumer
p7/l169-170 – approachs, dependance, cofusion, ariability
p9/l240 – erronrously
p15/l393 – overestimatons

**Reactions from quick review**

1) *In drawing their conclusions for potential energy of disturbances the authors should be cautious about the fact that for the possibility of Ep to be linked with the total wave energy (the reason why it is used in GW observations) it has to be computed from saturated spectra (e.g. VanZandt, 1985; Sacha et al., 2015). For saturated spectra, the dominant mode is naturally related to the largest vertical wavelength allowed for the analysis. What would be your conclusions if the GW ensemble (that you assume) would comprise ideally saturated spectra?*

Authors' response: As the reviewer states, from a measured spectra, the dominant mode is naturally related to the largest vertical wavelength allowed according to the selected vertical altitude interval. What we claim here is that any measured spectrum is obtained from a measured profile, and this will be unavoidably affected by the distortions above described due to the slanted nature of the sounding itself. Exceptionally, individual T vertical soundings as provided by lidar measurements, or from T, u, v or w, by radiosoundings under zero bakground wind conditions, will properly provide real GW spectra.

2) *In close relation with the spectral model you are using (5), it would also mean that all propagation directions are included. As you already mention in the text - e.g. the high aspect ratio expected in the Andes region due to a one dominant source of GW activity – the ideal spectra is far from the reach. But then the Ep should be used with caution already from the underlying theory. Note that your conclusions regarding the pseudo-momentum flux of Ern et al. (2004) should not be touched by this reviewer's concern as it is a conserved quantity.*

Authors' response: We agree that the peseudomomentum flux equation may be used as given by Ern et al, with the corresponding care discussed in the text in the calculation of the specific mean potential energy and considering the real vertical and horizontal wavelengths corresponding to the revailing GW mode. These wavelegths should be obtained after taking into consideration the expected distortions above described.

---

## Author Comment (AC1) · 28 Sep 2017

We acknowledge the comments and suggestions made by Dr. P. Pisoft (reviewer).

In particular regarding his suggestion in the Specific Comments Section, we feel that due to the complexity of any GW ensemble composed by monochromatic modes and the fact that our analysis should ideally be applied to each individual component, we are at this point able only to include an additional paragraph at lines 410-418 (see attached file):

" Up to now, from the satellite data at disposal, an attempt to quantitatively illustrate

the implications and possible misrepresentation (or distortion) of our general under-standing of GW parameters values from slanted soundings, as their global distribution and variability, seems unrealistic. After some research to improve this scenario, we are now working on previous GW parameters solution schemes which were modified for the use of close sounding groups of RO profiles. The method is currently being applied to calculate GW propagation direction, net MF and real vertical and horizontal wavelength for some case studies. The unavoidable constraint imposed to extend pre-liminay results to a GW climatological useful description is strictly conditioned by the still largely insufficient density of satellite-based soundings."

We feel that any further inference would result ambiguous and not straightforwardly applicable. GW parameters from low, medium and high frequency components suffer different distortions depending on their aspect ratio and sounding direction imposed by the respective radio occultation events. In particular, in the reference made to the status of our present work, we mention our attempt to go in depth over the open issues discussed in the paper.

The indicated typing errors were corrected.

Please also note the supplement to this comment:
https://www.atmos-meas-tech-discuss.net/amt-2017-192/amt-2017-192-AC1-supplement.pdf

———————————————————

[Figure]

**Supplement:**

[revised manuscript text omitted]

**Answer to:**
In particular regarding his suggestion in the Specific Comments Section, we feel that due to the complexity of any GW ensemble composed by monochromatic modes and the fact that our analysis should ideally be applied to each individual component, we are at this point able only to include an additional paragraph at lines 410-418. We feel that any further inference would result ambiguous and not straightforwardly applicable. GW parameters from low, medium and high frequency components suffer different distortions depending on their aspect ratio and sounding direction imposed by the respective radio occultation events. In particular, in the reference made to the status of our present work, we mention our attempt to go in depth over the open issues discussed in the paper.

The indicated typing errors were corrected.

---

## Referee Comment (RC2) · Anonymous Referee #2 · 31 Oct 2017

Review comments of "On the distortions in calculated GW parameters during slanted atmospheric soundings" by A. de la Torre et al.

This manuscript thoroughly investigated the "distortion factor" (D), namely the ratio between the apparent and the true vertical wavelengths, which is introduced by the slantwise observation. D is basically determined by the slantwise angle of the observing line of sight, and the wave propagation direction. The author also argued that the derived energy density (Ep) as well as the momentum flux (MF) are also exposed to the distortion due to the same reason, but quantitative assessment on this point is not established other than limited discussion on a single case study.

[Figure]

Two major slantwise observing techniques are studied: GPS-RO and SABER limb sounding. The main point was made on that, although both under-estimation (D<1) and over-estimation (D>1) can occur for both observing techniques, GPS-RO has much wider "observational window" to "see" many GWs, while SABER, due to its limb-sounding design, is only capable to observe only GWs that are aligned in a "correct" angle.

This research is carefully designed and conducted. Although similar topic has been mentioned and discussed in previous literatures, this is the first study I know to my best knowledge that focuses on studying this distortion factor introduced by slantwise sounding. Also, the point made on the two representative slantwise techniques (GPS-RO and SABER) is also novel and solid. I support eventual publication of this paper on AMT.

However, since I'm also one of the reviewers of the original manuscript before publication on AMTD, I still have some major comments to point out, as I don't see these points have been addressed properly before publication on AMTD. My major point is that: how do you quantify the association of "D" and the distortion of Ep and MF? The entire Section 3, although relate to another very important question (i.e., how slantwise sounding impacts/biases the Ep and MF estimation), is not quantitatively tied to the rest of this manuscript. Even for the single case study (Line 225-245), only one factor of the MF bias is actually estimated, while the other factor that biases Ep is only pointed out by Equation (9) but the exact value is not estimated. Besides, both factors are not apparently related to "D" that is discussed all through the rest of this paper.

If the authors can explicitly write "D" into Equation (8) and/or (9), please do so. Otherwise, I suggest remove Section 3 and only gently discuss the point that slantwise sounding would also distort the estimation of MF and Ep.

Another major point I have is that whether the distortion of "D" on a single measurement would be averaged out or not in the climatology? For example, with so many GPS-

RO soundings globally in one month of observation, would over-estimation and under-estimation of D likely not cause any distortion of the estimation of the climatology of the true vertical wavelength? Would the distortion still occur at certain regions where convective or topographic GWs dominate the local GW spectrum? Since we don't have the "truth", I suggest the author consider the following two strategies: (1) take high-resolution ECMWF or MIROC analysis and perform high-pass filtering to extract the "resolved GWs", and then apply GPS-RO and SABER viewing path to these GWs to construct a global "D" dataset; or (2) construct some very idealized GW spectra to represent convective, mountain and jet source generated GWs, and apply the same viewing path to construct an idealized global "D" spectrum. The former may take more effort and may be considered as future work. The latter is expected to be relatively easy. For example, you can take the spectrum suggested in: Alexander, M. J., and R. A. Vincent (2000), Gravity waves in the tropical lower stratosphere: A model study of seasonal and interannual variability, J. Geophys. Res., 105(D14), 17,983–17,993, doi:10.1029/2000JD900197. Or Gong, J., M. A. Geller, and L. Wang (2008), Source spectra information derived from U.S. high-resolution radiosonde data, J. Geophys. Res., 113, D10106, doi:10.1029/2007JD009252.

Minor points: Line 116: not sure what you mean. All in all, the current fasion of GWD parameterization is still heavily tuned and pretty "add-hoc" that lacks physical basis or observational constraints. You may add some references here to support your statement.

Figure 3: I suppose the color in the right panel corresponds to the magnitude of D? If yes, please make sure the left and right panels have consistent color scales. Right now it seems to me that the red color in the left panel has nothing to do with D.

Equation (4): Overbar by convention corresponds to "temporal" averaging. I agree with you that many "observational" GW study used spatial averaging to substitute this "temporal" averaging as the observations are transient. But you didn't make it clear in the context that the meaning of overbar is not "some" averaging but purely temporal

averaging.

Line 178: I don't quite understand here: Ep is proportional to the wave amplitude (i.e., T_hat), but I don't think it's necessarily proportional to the GWs with the largest wavelengths.

Line 196: longer -> larger.

Line 225: It would be better to explicitly add a sentence here stating that another factor that would distort MF is the distortion of Ep.

Line 233: From my impression, I think such a short horizontal wavelength GW is not suitable for GPS-RO to actually detect?

Equation (9): I did not read the companion paper by Hierro et al. (2017), so cannot judge about this specific case study: can you actually calculate how much Equation (9) is distorted for the dominant GW modes for this case?

Again, for the entire Section 3, I'd say it's not closely related to the rest of the paper. Line 278: add "that" after "mind". Also, what does "dispersion" exactly mean here?

Figure 5: Are you using the same bin size for both 5(a) and 5(b)? It would be better to add the "total = XXX" on the panel so readers would get a straightforward information of how different the sampling frequency is globally. Also, does the alpha-angle for GPS-RO vary with latitude significantly?

---

## Author Comment (AC2) · 22 Nov 2017

"On the distortions in calculated GW parameters during slanted atmospheric soundings" by A. de la Torre et al. We acknowledge and appreciate the details comments.

This manuscript thoroughly investigated the "distortion factor" (D), namely the ratio between the apparent and the true vertical wavelengths, which is introduced by the slantwise observation. D is basically determined by the slantwise angle of the observing line of sight, and the wave propagation direction. The author also argued that the derived energy density (Ep) as well as the momentum flux (MF) are also exposed to the distortion due to the same reason, but quantitative assessment on this point is not

established other than limited discussion on a single case study. Two major slantwise observing techniques are studied: GPS-RO and SABER limb sounding. The main point was made on that, although both under-estimation (D1) can occur for both observing techniques, GPS-RO has much wider "observational window" to "see" many GWs, while SABER, due to its limbsounding design, is only capable to observe only GWs that are aligned in a "correct" angle. This research is carefully designed and conducted. Although similar topic has been mentioned and discussed in previous literatures, this is the first study I know to my best knowledge that focuses on studying this distortion factor introduced by slantwise sounding. Also, the point made on the two representative slantwise techniques (GPSRO and SABER) is also novel and solid. I support eventual publication of this paper on AMT. However, since I'm also one of the reviewers of the original manuscript before publication on AMTD, I still have some major comments to point out, as I don't see these points have been addressed properly before publication on AMTD. My major point is that: how do you quantify the association of "D" and the distortion of Ep and MF? The entire Section 3, although relate to another very important question (i.e., how slantwise sounding impacts/biases the Ep and MF estimation), is not quantitatively tied to the rest of this manuscript. Even for the single case study (Line 225-245), only one factor of the MF bias is actually estimated, while the other factor that biases Ep is only pointed out by Equation (9) but the exact value is not estimated. Besides, both factors are not apparently related to "D" that is discussed all through the rest of this paper. If the authors can explicitly write "D" into Equation (8) and/or (9), please do so. Otherwise, I suggest remove Section 3 and only gently discuss the point that slantwise sounding would also distort the estimation of MF and Ep.

Section 3 has been completely written again, between pages 7 and 12 of the new version and several equations and discussion were included. We hope that these points have been more consistently addressed now. The main changes are highlighted in red.
Another major point I have is that whether the distortion of "D" on a single measurement would be averaged out or not in the climatology? For example, with so many GPS RO soundings globally in one month of observation, would over-estimation and underestimation of D likely not cause any distortion of the estimation of the climatology of the true vertical wavelength? Would the distortion still occur at certain regions where convective or topographic GWs dominate the local GW spectrum? Since we don't have the "truth", I suggest the author consider the following two strategies: (1) take high-resolution ECMWF or MIROC analysis and perform high-pass filtering to extract the "resolved GWs", and then apply GPS-RO and SABER viewing path to these GWs to construct a global "D" dataset; or (2) construct some very idealized GW spectra to represent convective, mountain and jet source generated GWs, and apply the same viewing path to construct an idealized global "D" spectrum. The former may take more effort and may be considered as future work. The latter is expected to be relatively easy. For example, you can take the spectrum suggested in: Alexander, M. J., and R. A. Vincent (2000), Gravity waves in the tropical lower stratosphere: A model study of seasonal and interannual variability, J. Geophys. Res., 105(D14), 17,983–17,993, doi:10.1029/2000JD900197. Or Gong, J., M. A. Geller, and L. Wang (2008), Source spectra information derived from U.S. high-resolution radiosonde data, J. Geophys. Res., 113, D10106, doi:10.1029/2007JD009252.

Here we followed the second suggestion by the reviewer. We feel that we reasonably improved the discussion now, taken into account the constraints of a one dimensional model. We mainly tried to provide a good idea of the possible application and limitations of our results to past and future slanted atmospheric soundings.

Minor points: Line 116: not sure what you mean. All in all, the current fasion of GWD parameterization is still heavily tuned and pretty "add-hoc" that lacks physical basis or observational constraints. You may add some references here to support your statement.

A sentence and a reference from a review were added in lines 116-119.

Figure 3: I suppose the color in the right panel corresponds to the magnitude of D? If yes, please make sure the left and right panels have consistent color scales. Right now it seems to me that the red color in the left panel has nothing to do with D.

This is now more clear with the color bar and added comment in the caption.

Equation (4): Overbar by convention corresponds to "temporal" averaging. I agree with you that many "observational" GW study used spatial averaging to substitute this "temporal" averaging as the observations are transient. But you didn't make it clear in the context that the meaning of overbar is not "some" averaging but purely temporal C3 averaging. Line 178: I don't quite understand here: Ep is proportional to the wave amplitude (i.e., T_hat), but I don't think it's necessarily proportional to the GWs with the largest wavelengths.

I hope that these point are more clear between lines 182 and 184.

Line 196: longer -> larger.

Done.

Line 225: It would be better to explicitly add a sentence here stating that another factor that would distort MF is the distortion of Ep. Line 233: From my impression, I think such a short horizontal wavelength GW is not suitable for GPS-RO to actually detect? Equation (9): I did not read the companion paper by Hierro et al. (2017), so cannot judge about this specific case study: can you actually calculate how much Equation (9) is distorted for the dominant GW modes for this case? Again, for the entire Section 3, I'd say it's not closely related to the rest of the paper.

We feel that these and other additional points are more discussed and detailed in the new Section 3.

Line 278: add "that" after "mind". Also, what does "dispersion" exactly mean here?

Done. We changed "dispersion" by "distribution" which better express the idea that we

had at this point.

Figure 5: Are you using the same bin size for both 5(a) and 5(b)? It would be better to add the "total = XXX" on the panel so readers would get a straightforward information of how different the sampling frequency is globally.

A comment was included in the caption to the figure with the exact number of soundings.

Also, does the alpha-angle for GPS-RO vary with latitude significantly?

No significant variation with latitude was detected.

Please also note the supplement to this comment:
https://www.atmos-meas-tech-discuss.net/amt-2017-192/amt-2017-192-AC2-supplement.zip
* * *